# Attitudes Toward Patient Safety among Medical Students in Malaysia

**DOI:** 10.3390/ijerph17217721

**Published:** 2020-10-22

**Authors:** Sathia Prakash Nadarajan, Sumitra Ropini Karuthan, Jeevitha Rajasingam, Karuthan Chinna

**Affiliations:** 1Department of Social and Preventive Medicine, Faculty of Medicine, University Malaya, Kuala Lumpur 50603, Malaysia; drsp@spcaregroup.com; 2Ministry of Health Malaysia, Seremban 70300, Malaysia; sumiruby94@gmail.com; 3Medical Education Research and Development Unit, Faculty of Medicine, University Malaya, Kuala Lumpur 50603, Malaysia; jeevitharajasingam@gmail.com; 4School of Medicine, Faculty of Health and Medical Sciences, Taylors’s University, Subang Jaya 47500, Malaysia

**Keywords:** patient safety, medical students, attitudes

## Abstract

The biggest challenge in moving toward a safer healthcare system is patient safety culture—that is, the prevention of harm to patients. Safe medical practices can prevent doing harm to the patients. For this, healthcare professionals must have good attitudes toward patient safety. Medical education plays an important role in promoting patient safety and patient safety attitudes. A study was conducted among medical students in Malaysia to assess their perceptions toward patient safety, using the 26-items Attitudes Toward Patient Safety Questionnaire (APSQ-III). In the analysis, the average percentage of positive responses (APPR) were computed for each domain, and APPR values of ≥75 were used as an indicator of positive perception. Out of the nine domains of APSQ, the students’ attitude was positive in six—Safety Training (85.2%), Error Reporting (76.3%), Working Hours (89.5%), Error Inevitability (86.1%), Team Functioning (94.6%), and Patient Involvement (80.1%). The desired level of positive attitude was not met in Disclosure Responsibility (68.5%), Professional Incompetence (70.0%), and Safety Curriculum (71.1%). APRR for disclosure responsibility was high among the first-year students, but, generally, the effect wore off over the years of study. The results support the need to enhance perception on Disclosure Responsibility, Professional Incompetence, and Safety Curriculum among the medical students in Malaysia.

## 1. Introduction

The first World Patient Safety Day was commemorated on 17 September 2019 with the theme “Patient Safety: a global health priority”. The emphasis was on openness and “blame-free” environments as the minimum conditions for enacting a safety culture. World Patient Safety Day is a prompt to everyone involved in healthcare on the seriousness of patient safety. Worldwide, every day, countless patients are put at risk by unsafe care and end up with adverse events caused by the very system that was supposed to help them get better. Based on World Health Organization (WHO) estimates, one in every 10 patients is harmed while receiving hospital care; the risk of patient death occurring due to a preventable medical accident, while receiving healthcare, is estimated to be one in 300; and 15% of all hospital expenses are incurred as a result of treating failures in patient safety [1]. Unsafe care is likely one of the 10 leading causes of death and disability across the world [1]. Specifically, in the low-and middle-income countries (LMICs), about 134 million adverse events occur each year due to unsafe care in hospitals, resulting in 2.6 million deaths annually [2]. This raises the question, “How safe are our patients?”

Patient safety, as defined by the Institute of Medicine (IOM) of the National Academy of Sciences, is the prevention of harm to patients [3]. Based on the 1999 IOM report’s main conclusions, majority of medical errors do not result from individual recklessness or the actions of a particular group, but more commonly are caused by faulty systems, processes, and conditions that lead people to make mistakes or fail to prevent them [4]. In patient safety, the emphasis is on the system of care delivery that is supposed to prevent errors, learn from the errors that do occur, and is based on a culture of safety that involves health care professionals, organizations, and patients [5]. Shreve et al. (2010) defined medical error as an act that produces a preventable adverse outcome compared to the natural progression of disease that leads to injury or death [6]. More specifically, Carruthers et al. (2009) defined medical error as “the failure to properly carry out an appropriately planned action (slip) or successfully carrying out an incorrect action (mistake) where there is potential for patient harm” [7].

Patient safety attitude refers to the shared attitudes, beliefs, values, and assumptions that underlie how people perceive and act upon safety issues within their organization [8]. The components are concerned with the healthcare providers’ attitudes about the organization, the perceived work environment, perceived managerial support, and perceived teamwork and supervision [9]. The major elements of patient safety science include voicing out safety issues without fear of retribution, having feedback mechanisms for continuous improvements, and practicing transparency in safety culture [10]. The biggest challenge to moving toward a safer healthcare system is changing the culture from one of blaming individuals for errors to one in which errors are treated not as personal failures but as opportunities to improve the system and prevent harm [1].

Medical school education plays an important role in promoting patient safety. Although patient safety knowledge is essential, it is also crucial that patient safety attitudes are evaluated as attitudes can considerably influence behaviors [10]. Assessment of attitude is a reliable measure for evaluating the efficacy of patient safety programs [11]. Medical students’ experiences during clinical rotations can influence their attitudes toward patient safety and their future behavior. While positive experiences enhance patient safety, negative experiences may lead to more harm to the patients. Even though many researches had been conducted elsewhere on medical students’ attitudes towards patient safety [11,12,13,14,15,16,17,18], such a study is yet to be conducted in Malaysia. Studies conducted in the USA, Pakistan, Korea, Singapore, and Hong Kong have shown inadequate awareness of medical error reporting. Studies have also stated a lack of or incomplete patient safety curriculum in medical schools. 

In Malaysia, in the year 2017, there was a remarkable increase in the number of patient safety incidents reported via electronic Incidence Reporting (e-IR) [19]. In 2017, there were 5689 reported incidents compared to the 2016 figure of 2769. This could mean that either the number of incidents has increased two-fold in one year or the reporting rate increased. However, it is not certain how many cases go unreported. It is not certain what actions were taken, what lessons were learnt, and what strategies were implemented to avoid such future incidences. 

## 2. Materials and Methods

A cross-sectional study was conducted at a premier medical university in the city of Kuala Lumpur, Malaysia, in January 2020. The study was approved by the university’s ethic committee. At the time of this study, there were 700 students registered at the faculty of medicine. Taking proportion of positive perception for the domains as 50%, confidence level of 95%, and 5% a margin of error, the minimum required sample size was 250 [20]. 

A self-administered questionnaire prepared in Google Form was sent to all the students. An attached cover letter explained the purpose of the study and the students were informed that participation was voluntary. They were assured that information provided by them will be kept confidential and only used for the purpose of this study. Respondents’ consents were taken before they could proceed with the survey.

In this study, the 26-item Attitudes to Patient Safety Questionnaire (APSQ-III) developed by Carruthers et al. in 2009 [7] was used. This questionnaire was designed to assess medical students’ attitude in 9 safety domains. The internal consistency of the items in the nine domains ranged between 0.64 and 0.82.

The nine dimensions of patient safety attitude and the items are
(1)Patient safety training received (ST);(2)Error reporting confidence (ER);(3)Working hours as an error cause (WH);(4)Error inevitability (EI);(5)Professional incompetence as an error cause (PI);(6)Disclosure responsibility (DR);(7)Team functioning (TF);(8)Patient involvement in reducing error (PI);(9)Importance of patient safety in the curriculum (SC).

Even though Carruthers et al. used a Likert scale of 1–7, some studies have used the Likert scale of 1–5 [17,18]. In this study, the items were measured on a scale of 1–5, where a response of 1 indicated disagreement while a response of 5 indicated agreement to the statement. APSQ-III has been validated in Malaysia [21]. In this study, test of internal consistency, based on a pilot study among 30 students, yielded Cronbach’s alpha values between 0.72 and 0.85 for the nine constructs. In addition to the APSQ-III items, questions on students’ gender and year of study were added in the questionnaire.

In this study, the average percentage of positive responses (APPR), defined as the average of the item-level percent positive responses within an APSQ dimension, was used to measure the level of positive attitude towards patient safety among the medical students [18]. For this, first, the responses for the items in APSQ were dichotomized into 1 (positive) or 0 (negative). For items 11, 13, 14, 15, 16, 17, and 25, responses of 1 or 2 were coded as positive and for other items, the responses of 4 or 5 were coded as positive. APRR for each domain was then calculated as the average percentage of the positive perception statements [18]. On deciding the desirable level of positive attitude in a domain, Nordén-Hägg et al. (2010) recommended a stringent cut-off point of ≥75 out of 100 [22]. In this study we adopted the cut-off point of ≥75 to calculate and chart APPR for the nine APSQ domains.

IBM SPSS Statistics for Windows, Version 24.0 software (IBM Corporation, Armonk, NY, USA) was used in the data analysis. Categorical variables, such as gender, year of study, and responses to statements, were described as frequencies and proportions. APRRs were presented as means and standard deviations. In the analyses, to test the differences in APRRs for the nine domains, the multivariate general linear model (GLM) was used. For all tests, the level of significance was set as 0.05.

## 3. Results

The questionnaire was sent electronically to all 700 students registered at the faculty. Out of these, 324 responses were received. The response rate was 46.3%. The majority (63.0%) of the respondents were females, and the distribution of the students by year of study was quite proportionate (Table 1). 

The responses for the 26 statement in APSQ are presented in Table 2. The internal consistencies for the items in each domain ranged from 0.665 to 0.900. Item-wise, positive responses ranged from 63.6% to 94.8%; the lowest for “Better multi-disciplinary teamwork will reduce medical errors” (63.6%) and the highest for “Teaching teamwork skills will reduce medical errors” (94.8%). The positives responses were more than 75% for all the items in Safety Training, Working Hours, and Error Inevitability domains. Negative perception to the three items in Disclosure Responsibility domain was high, ranging from 22.2%, 24.1%, and 27.8%.

The average percentage of positive responses (APPRs), for the nine domains, overall and by year of study and gender, are presented in Table 3 and Figure 1. Overall, APRR was the highest for the Team Functioning domain (94.6%) and was the lowest for Disclosure Responsivity domain (68.5%). APRRs were below 75% for Profession Incompetence, Disclosure Responsivity, and Safety Curriculum. 

By year of study, there were significant differences in Error Reporting (*p* = 0.001), Professional Incompetence (*p* < 0001) and Disclosure Responsibility (*p* = 0.002) domains and between year of study; APPR for Error Reporting among third-year students was higher compared to the second-year students, APPR for Professional Incompetence among fifth-year students was higher compared to the first-year students, and APRR for Disclosure Responsibility among the first-year students was higher compared to the second-year students. There were significant differences in perceptions on Error Reporting (*p* = 0.002) and Professional Incompetence (*p* = 0.012) between gender; APRR for Error Reporting was higher among the males, and APRR for Professional Incompetence was higher among the females.

## 4. Discussion

This study investigated attitudes about patient safety among medical students in Malaysia. Among the nine domains of APSQ, the average positive response rate (APRR) was the highest for Team Functioning domain (94.6%), and it was the lowest for Disclosure Responsibility domain (68.5%). The attitude of patient safety among Malaysian medical students is above the acceptable level of 75% (Nordén-Hägg et al., 2010) in six out of the nine domains [21]. Overall, the students have poorer attitudes in Disclosure Responsibility, Professional Incompetence, and Safety Curriculum, with APRR values of 68.5%, 70%, and 71.1%, respectively.

Generally, Malaysian students perceived that their medical education has prepared them in recognizing and avoiding medical errors. Malaysian students’ have high positive perception in Error Reporting, Similar to students in other Asian countries, the medical students in Malaysia are concerned about working hours. They agree that the risk of making errors is high when one has to work long hours without sufficient breaks in between.

In this study more than 80% of the students were agreeable on Error Inevitability. This means the students are aware that human error is inevitable and that even an experienced practitioner can make errors. Only 70% of the Malaysia students showed positive perception on Professional Incompetence as a source of error. Student responses in the Professional Incompetence subscale indicates that students believe that individual failures are the primary root cause of errors. Based on the IOM report, majority of medical errors do not result from individual recklessness or the actions of a particular group, but more commonly are caused by systems, processes, and conditions that lead people to make mistakes or fail to prevent them. This concept underpins the interactions of justice, teamwork, and continuous quality improvement [23]. Even though most of the students agreed that human error is inevitable, they still seem to blame others for the occurrences of such error. This could mean that either “blame-culture” is still dominant in the culture or the students are not very sure of what constitutes medical errors. There is also some concern on whether the students were confused as the items in this domain are negatively worded compared to the items in other domains. Generally, the perception on Professional Incompetence among students in other countries was also low.

Perception on disclosure responsibility among Malaysia students was low (68.5%). Many medical students think that it is not necessary to report an error if it does not result in undesirable outcome and are not willing to disclose their responsibilities in such errors. This is a poor attitude as every error is a learning opportunity, no matter how trivial it is. Brian-storming such events helps the team members help each other’s experience. Providing detailed instructions on disclosure guidelines and improving communication skills could help the students to improve their attitude toward Reporting Responsibility [18]. Malaysian medical students’ perceptions on Team Functioning and Patient Involvement in medical care were high. It is good to note that the medical students are willing to work as a team and with people from other disciplines as well, rather than as individuals. Similarly, the fact that they would like the patients to be involved in their care is encouraging.

With respect to the Safety Curriculum, Malaysian students’ perception was not high Among the three items in this domain most of the Malaysian students had the idea that patient safety issues cannot be taught and can only be learned by clinical experience, when they qualified. In APSQ, this item is worded negatively compared to the other two. We are not sure if this has influenced the overall positive perception to be low. There were significant differences in the perception of Error Reporting, Professional Incompetence and Disclosure Responsibility domains between year of study and Error Reporting and Professional Incompetence domains between gender. Third-year students were more positive towards Error Reporting compared the second-year students. As the third-year students are in their clinical posting, they could be more comfortable in Error Reporting. The positive perception on Professional Incompetence being lower among the first-year students compared to those in their final year could be due to lack of ability among the freshman to recognize what constitutes an error. Even though the freshmen are very positive on Disclosure Responsibility, the effect generally wears off over the years of study. This could reflect a negative culture in the system, which is a cause for concern.

## 5. Conclusions

Generally, Malaysian medical students have a positive attitude toward patient safety. However, the students do not have the desired level of positive attitude in Professional Incompetence and Disclosure Responsibility. It appears there is some confusion among the medical students on what constitutes a medical error. The medical curriculum needs to address this area. To improve the students’ Disclosure Responsibility, the students should be provided with detailed instructions on how to make the disclosures and encouraged to take every error as a learning opportunity. The third-year medical students, who are in their clinical posting, fared better in most dimensions. The perception that long working long hours could lead to medical errors should be viewed seriously. Continuous updating of formal curriculum on patient safety and monitoring the students’ practice in clinical setting would help to create patient safety culture in clinical practice.

## 6. Limitations

First, as in all surveys using questionnaires, it is not possible to determine how honestly the students would have answered the questions. This could have biased the results. Second, this study was conducted at a premier medical university in the city of Kuala Lumpur. There are 32 medical universities in Malaysia (11 public and 21 private), with varying numbers of students. As such, the results of study may not be generalizable to all medical students in the country. We recommend future studies to include students from other universities as well.

## Figures and Tables

**Figure 1 ijerph-17-07721-f001:**
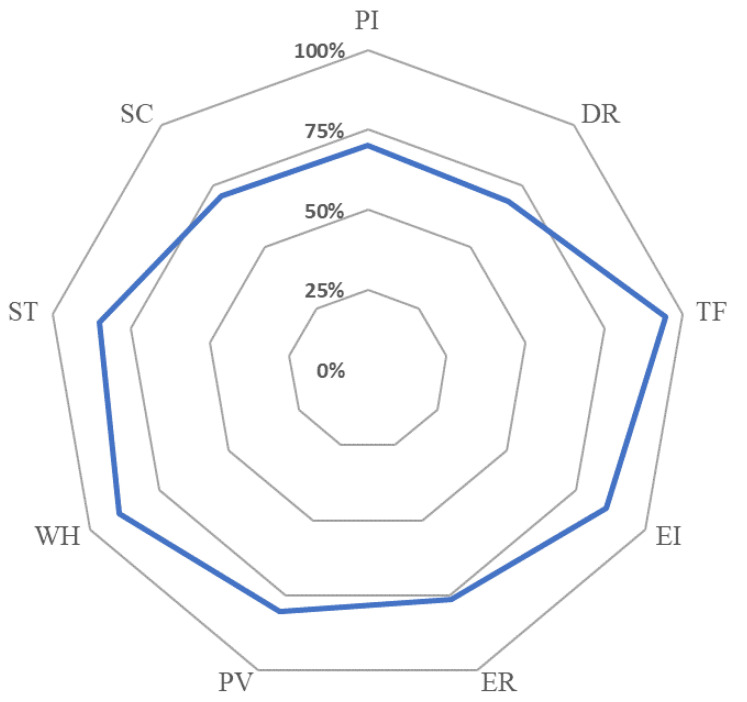
Average positive response rate (APRR) for the nine domains in APSQ.

**Table 1 ijerph-17-07721-t001:** Demographic characteristics of the respondents.

Variable	Frequency	Percentage
Gender		
Male	120	37.0%
Female	204	63.0%
Year of study		
1	60	18.5%
2	75	23.1%
3	69	21.3%
4	63	19.4%
5	57	17.6%

**Table 2 ijerph-17-07721-t002:** Responses to the safety attitude statements in the Attitudes Toward Patient Safety Questionnaire (APSQ-III).

Statements	Mean ± SD	Negative n (%)	Neutral n (%)	Positive n (%)
*Safety Training (Cronbach’s alpha = 0.832)*				
My training has prepared me to understand the causes of medical errors	4.1 ± 0.8	12 (3.7)	42 (13.0)	270 (83.3)
I have a good understanding of patient safety issues as a result of my undergraduate medical training	4.1 ± 0.7	6 (1.9)	27 (8.3)	291 (89.8)
My training has prepared me to prevent medical errors	4.1 ± 0.8	12 (3.7)	45 (13.89)	267 (82.4)
*Error Reporting (Cronbach’s alpha = 0.797)*				
I would feel comfortable reporting any errors I had made, no matter how serious the outcome had been for the patient	4.0 ± 0.8	21 (6.5)	50 (15.4)	253 (78.1)
I would feel comfortable reporting any errors other people had made, no matter how serious the outcome had been for the patient	3.8 ± 0.9	23 (7.1)	78 (24.1)	223 (68.8)
I am confident I could talk openly to my supervisor about an error I had made if it had resulted in potential or actual harm to my patient	4.1 ± 0.9	25 (7.7)	33 (10.2)	266 (82.1)
*Working Hours (Cronbach’s alpha = 0.830)*				
Shorter shifts for doctors will reduce medical errors	4.4 ± 0.9	15 (4.6)	39 (12.0)	270 (83.3)
By not taking regular breaks during shifts doctors are at an increased risk of making errors	4.5 ± 0.7	9 (2.8)	12 (3.7)	303 (93.5)
The number of hours doctors work increases the likelihood of making medical errors	4.0 ± 0.7	6 (1.9)	21 (6.5)	297 (91.7)
*Error Inevitability (Cronbach’s alpha = 0.812)*				
Even the most experienced and competent doctors make errors	4.5 ± 0.8	12 (3.7)	20 (6.2)	292 (90.1)
A true professional does not make mistakes or errors	1.7 ± 0.8	11 (3.4)	30 (9.3)	283 (87.3)
Human error is inevitable	4.2 ± 0.8	12 (3.7)	50 (15.4)	262 (80.9)
*Professional Incompetence* * (Cronbach’s alpha = 0.824)*				
Most medical errors result from careless nurses	2.3 ± 0.7	14 (4.3%)	78 (24.1)	232 (71.6)
If people paid more attention at work, medical errors would be avoided	2.3 ± 0.7	15 (4.6%)	75 (23.1)	234 (72.2)
Most medical errors result from careless doctors	2.4 ± 0.8	25 (7.7%)	85 (26.2)	214 (66.0)
Medical errors are a sign of incompetence	2.2 ± 0.8	21 (6.5%)	76 (23.5)	227 (70.1)
*Disclosure responsibility (Cronbach’s alpha = 0.900)*				
It is not necessary to report errors which do not result in adverse outcomes for the patient	2.5 ± 1.0	90 (27.8)	10 (3.1)	224 (69.1)
Doctors have a responsibility to disclose errors to patients only if they result in patient harm	3.6 ± 0.9	72 (22.2)	31 (9.6)	221 (68.2)
All medical errors should be reported.	3.6 ± 1.0	78 (24.1)	25 (7.7)	221 (68.2)
*Team Functioning (Cronbach’s alpha = 790)*				
Better multi-disciplinary teamwork will reduce medical errors	4.5 ± 0.7	1 (0.3)	117 (36.1)	206 (63.6)
Teaching teamwork skills will reduce medical errors	4.5 ± 0.6	0	17 (5.2)	307 (94.8)
*Patient Involvement (Cronbach’s alpha = 0.665)*				
Patients have an important role in preventing medical errors	4.1 ± 0.8	3 (0.9)	90 (27.8)	231 (71.3)
Encouraging patients to be more involved in their care can help to reduce the risk of medical errors occurring	4.3 ± 0.7	0	36 (11.1)	288 (88.9)
*Safety Curriculum (Cronbach’s alpha = 0.869)*				
Teaching students about patient safety should be an important priority in medical students training	4.1 ± 0.8	0	84 (25.9)	240 (74.1)
Patient safety issues cannot be taught and can only be learned by clinical experience when qualified	2.0 ± 0.8	0	101 (31.2)	223 (68.8)
Learning about patient safety issues before I qualify will enable me to become a more effective doctor	4.0 ± 0.8	0	96 (29.6)	228 (70.4)

**Table 3 ijerph-17-07721-t003:** Average positive response rate (APRR) the nine domains in APSQ.

	ST	ER * ^#^	WH	EI	PI * ^#^	DR *	TF	PV	SC
Overall	85.2	76.3	89.5	86.1	70.0	68.5	94.6	80.1	71.1
Year 1	81.7	72.2	93.3	83.3	59.6	81.7	92.5	85.0	75.0
Year 2	82.7	66.7	92.0	88.0	67.7	60.4	94.0	76.0	60.4
Year 3	84.1	86.5	87.0	82.1	68.1	75.4	100.0	89.1	66.7
Year 4	85.7	77.2	83.1	86.2	75.0	64.0	91.3	75.4	77.8
Year 5	93.0	80.1	92.4	91.2	80.7	62.0	94.7	74.6	78.9
Male	85.0	83.6	89.2	82.2	66.3	70.8	96.3	85.0	70.8
Female	85.3	72.1	89.7	88.4	72.2	67.2	93.6	77.2	71.2

* significant difference between year of study. ^#^ significant difference by gender.

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
