# Peer review of "Attitudes Toward Patient Safety among Medical Students in Malaysia"

_ijerph, 2020, doi:10.3390/ijerph17217721_

Round 1

Reviewer 1 Report

Dear Authors: 

Congratulations!.

Just review the number of references. 

I have only one observation. May be you can add some comments about reasons to select this survey an no others.

Regards,

Author Response

We thank you for reviewing our manuscript

We have made some ammendments in the manuscript

Thank you

Reviewer 2 Report

Overall, I thought this was an excellent study and an interesting addition to the field of patient safety. I think so much begins with the medical (or other healthcare) education, and being able to identify areas where improvement in the curriculum are needed is a really important step.

I think there are a few areas that could be strengthened with some additional clarification. In the beginning of the abstract (the first 3 sentences), the phrase "patient safety" is used without a definition, and because this can mean many different things to different audiences, I think you should be specific, even with a few words of a definition as you have given in the introduction. 

In the introduction, lines 34-35, the line you are referencing from WHO ("In comparison, the risk of patient death occurring due to a preventable medical accident, while receiving health care, is estimated to be 1 in 300.") is different that what you are stating in your manuscript ("one in every 300 patients die due to preventable adverse events"). Medical accidents (sometimes also called medical errors) are not always the same as adverse events, so I would either suggest using the language used by WHO. For your reference to Shreve, lines 47-49, the authors are actually defining "medical error" and not simply "error" - based on the title of the paper as well as the other information in their intro - so I suggest being more specific there. In the last paragraph of the introduction, line 74, you mention "e-IR" but do not explain what this is, and I think it would be helpful for the reader and give additional context for your rationale for your study.

Table 2 - is the plus/minus "s" in the heading supposed to be plus/minus "sd" (as in standard deviation)? This should be clear so as not to confuse the reader (with something else like standard error of the mean).

In the results, line 155, is "significant differences" referring to the statistical sense? If so, the relevant scores and p values should be included.

In the discussion, lines 221-224, I would omit unless you can provide support for these assumptions of male versus female qualities, such as a study or another source.

Author Response

Firt of all, we thank you for your valuable comments

We are attaching our responses to your comments

Than you

Reviewer 3 Report

Thank you for an opportunity to review your article.  My recommendation will focus on your discussion section, as I don't find significant concerns in other sections except for:

  • I presume invitation to participate was sent to all 700 students, but it may be worth mentioning specific number;  any studies using voluntary surveys should specify response rate to survey request clearly.

The big concern lies in your discussion section. 

This was a single-site cross sectional study performed at "a premier medical university in the city of KL." Your discussion extrapolated the outcome of this study to represent all Malaysian medical students, which I don't think is appropriate based on a single site study as there could be significant variability based on which medical school and which city the school is located. 

Your discussion focuses on comparison of Malaysian medical students with other nations, but those studies also have limited scope, and should NOT be used to make grandeur statement of comparing nations to another. 

You also fail to provide limitations of this study. 

Finally, your reference page starts with number 121.  Please fix this. 

Author Response

First of all, thank you for your valuable comments

We are attaching our responses to your comments

Thank you

Round 2

Reviewer 3 Report

Thank you for your edition.  

I still have one major issue and a minor editing issue.

You did not change your discussion section.  As I stated, making comparison with other nations based on a survey (with <50% response rate) from one institution is completely inappropriate.  Other studies you have cited for comparison also have limited scope and should not be used to declare safety culture of an entire nation.   

Author Response

We thank you for your comments. We have orrected the discussion section as below..

  1. Discussion

This study investigated attitudes about patient safety among medical students in Malaysia. Among the 9 domains of APSQ, the average positive response rate (APRR) was the highest for Team Functioning domain (94.6%) and it was the lowest for Disclosure Responsibility domain (68.5%).  The attitude of patient safety among Malaysian medical students is above the acceptable level of 75% (Nordén-Hägg et al, 2010) in  6 out of the 9 domains [21] Overall, the students have poorer attitude in  Disclosure Responsibility, Professional Incompetence and Safety Curriculum, with APRR values of 68.5%, 70% and 71.1%, respectively.

Generally, Malaysian students perceived their medical education has prepared them in recognizing and avoiding medical errors. Malaysian students’ have high positive perception in Error Reporting, Similar to students in other Asian countries, the medical students in Malaysia are concerned about working hours. They agree that the risk of making errors is high when one has to work long hours without sufficient breaks in between.

In this study more than 80% of the students were agreeable on Error Inevitability. This means the students are aware that human error is inevitable and that even an experienced practitioner can make errors. Only 70% of the Malaysia students showed positive perception on Professional Incompetence as a source of error. Student responses in the Professional Incompetence subscale indicates that students believe that individual failures are the primary root cause of errors. Based IOM report, majority of medical errors do not result from individual recklessness or the actions of a particular group, but more commonly are caused by systems, processes, and conditions that lead people to make mistakes or fail to prevent them. This concept underpins the interactions of justice, teamwork, and continuous quality improvement [23] Even though most of the students agreed that human error is inevitable, they still seem to blame others for the occurrences of such error. This could mean that either the ‘blame-culture’ is still dominant in the culture or the students are not very sure of what constitutes medical errors. There is also some concern on whether the students were confused as the items in this domain are negatively worded compared to the items in other domains.  Generally, the perception on Professional Incompetence among students in other countries was also low.

Perception on disclosure responsibility among Malaysia students was low (68.5%). Many medical students think that it is not necessary to report an error if it does not result in undesirable outcome and are not willing to disclose their responsibilities in such errors. This is a poor attitude as every error is a learning opportunity, no matter how trivial it is. Brian-storming such events helps the team members help each other’s experience. Providing detailed instructions on disclosure guidelines and improving communication skills could help the students to improve their attitude towards Reporting Responsibility [18] Malaysian medical students’ perceptions on Team Functioning and Patient Involvement in medical care were high. It is good to note that the medical students are willing to work as team and with people from other disciplines as well, rather than as individuals. Similarly, the fact that they would like the patients to be involved in their care is encouraging  

With respect to the Safety Curriculum, Malaysian students’ perception was not high Among the three items in this domain most of the Malaysian students had the idea that patient safety issues cannot be taught and can only be learned by clinical experience, when they qualified. In APSQ, this item is worded negatively compared to the other two. We are not sure if this has influenced the overall positive perception to be low. There were significant differences in the perception of Error Reporting, Professional Incompetence and Disclosure Responsibility domains between year of study and Error Reporting and Professional Incompetence domains between gender. Third-year students were more positive towards Error Reporting compared the second-year students. As the third-year students are in their clinical posting, they could be more comfortable in Error Reporting.  The positive perception on Professional Incompetence being lower among the first-year students compared to those in their final year could be due to lack of ability among the freshman to recognize what constitutes an error.  Even though the freshmen are very positive on Disclosure Responsibility, the effect generally wears off over the years of study. This could reflect a negative culture in the system, which is a cause for concern.

Thank you